# Exploring the Associated Genetic Causes of Diabetic Retinopathy as a Model of Inflammation in Retinal Diseases

**DOI:** 10.3390/ijms25105456

**Published:** 2024-05-17

**Authors:** Francesco Cappellani, Carl D. Regillo, Julia A. Haller, Caterina Gagliano, Jose S. Pulido

**Affiliations:** 1Wills Eye Hospital, Thomas Jefferson University, Philadelphia, PA 19107, USA; francescocappellani@hotmail.com (F.C.);; 2Department of Ophthalmology, University of Catania, 95123 Catania, Italy; 3Faculty of Medicine and Surgery, University of Enna Kore, 94100 Enna, Italy; caterina_gagliano@hotmail.com; 4Ocular Immunology and Rare Diseases Unit, San Marco Hospital, 95123 Catania, Italy

**Keywords:** diabetic retinopathy, proteomics, transcriptomics, immune system, inflammation, complement system, OmicsPred, omics, advanced glycosylation end-product receptor

## Abstract

To investigate potential biomarkers and biological processes associated with diabetic retinopathy (DR) using transcriptomic and proteomic data. The OmicsPred PheWAS application was interrogated to identify genes and proteins associated with DR and diabetes mellitus (DM) at a false discovery rate (FDR)-adjusted *p*-value of <0.05 and also <0.005. Gene Ontology PANTHER analysis and STRING database analysis were conducted to explore the biological processes and protein interactions related to the identified biomarkers. The interrogation identified 49 genes and 22 proteins associated with DR and/or DM; these were divided into those uniquely associated with diabetic retinopathy, uniquely associated with diabetes mellitus, and the ones seen in both conditions. The Gene Ontology PANTHER and STRING database analyses highlighted associations of several genes and proteins associated with diabetic retinopathy with adaptive immune response, valyl-TRNA aminoacylation, complement activation, and immune system processes. Our analyses highlight potential transcriptomic and proteomic biomarkers for DR and emphasize the association of known aspects of immune response, the complement system, advanced glycosylation end-product formation, and specific receptor and mitochondrial function with DR pathophysiology. These findings may suggest pathways for future research into novel diagnostic and therapeutic strategies for DR.

## 1. Introduction

Diabetic retinopathy (DR) is a complication of diabetes mellitus, affecting millions of people worldwide. Recent estimates indicate that there were approximately 9.6 million individuals living with diabetic retinopathy in the United States in 2021, of which 1.84 million were suffering from vision-threatening stages of the disease [1]. DR is characterized by progressive retinal damage, which can lead to vision loss and irreversible blindness if left untreated. The early detection and management of DR are crucial to prevent or delay the progression of the disease. The pathogenesis of DR is complex and multifactorial, involving metabolic, genetic, and environmental factors [2]. A better understanding of the molecular mechanisms underlying DR is essential for the development of novel diagnostic and therapeutic strategies. Omics modalities, such as genomics, transcriptomics, proteomics, and metabolomics, are useful tools to investigate and analyze the underlying biology of diseases and traits [3]. A recent study examined a large cohort from the INTERVAL study with extensive multi-omics data and used machine learning to train genetic scores for 17,227 molecular traits [4]. These genetic scores were evaluated through external validation across cohorts of individuals with European, Asian, and African American ancestries. The authors developed the OmicsPred database (https://www.omicspred.org/, accessed on 3 May 2023), an online portal that provides access to all genetic scores, validation results, and biomarker analyses, serving as an open and updatable resource. In addition, the researchers conducted a phenome-wide association analysis (PheWAS) using PheCodes, using a large cohort of individuals of European-only ancestry from the UK Biobank (UKB) [4,5]. OmicsPred also hosts descriptions and summary results from their phenome-wide association analysis.

In this study, we interrogated the OmicsPred database for potential biomarkers for DR, which might contribute to a better understanding of the disease’s underlying pathophysiology and guide preventive or therapeutic strategies. Then, we explored these biomarkers further by correlating them with potentially revelatory pathological pathways for DR using Gene Ontology PANTHER analysis (Protein ANalysis THrough Evolutionary Relationships; http://www.pantherdb.org/, accessed on 15 May 2023) [6] and the STRING database (https://string-db.org/, accessed on 18 May 2023) [7].

## 2. Results

### 2.1. Proteomic and Transcriptomic Associations with DR and DM

Using the OmicsPred PheWAS application, we interrogated the following two conditions: diabetic retinopathy (DR) (Phecode 250.7) and diabetes mellitus (DM) (Phecode 250). In the transcriptomic database, there were 49 genes associated with DR and/or DM. In the proteomic database, 22 proteins were associated with DR and/or DM (Table 1a,b).

The genes and proteins were then divided by FDR-adjusted *p*-value (with two different thresholds (<0.05 and <0.005)) and into three groups: those uniquely associated with diabetic retinopathy, those uniquely associated with diabetes mellitus, and the ones in common between these two conditions. A total of 23 genes and 10 proteins were found to be uniquely associated with diabetic retinopathy, with an FDR-adjusted *p*-value threshold <0.05, reducing to 10 genes and 7 proteins, respectively, when the FDR-adjusted *p*-value threshold was lowered to <0.005. Meanwhile, 16 genes and 11 proteins were found to be in common with both conditions (FDR-adjusted *p*-value < 0.05), decreasing to 14 genes and 5 proteins by lowering the threshold to <0.005. Finally, 10 genes and 1 protein were found to be uniquely associated with diabetes mellitus (FDR-adjusted *p*-value < 0.05), and 4 genes and 1 protein by lowering the threshold to <0.005. Two nonprotein-coding genes (*LINC00243* and *AL662795.2*) were found to be uniquely associated with diabetic retinopathy; one pseudogene was found to be in common with both conditions (*AL645933.4*). This was best captured by Venn diagrams (Figure 1a,b).

Interestingly, our analyses highlighted the associations at both the transcriptomic and proteomic levels of *C2* and *C4a*, as both the genes encoding these proteins and the proteins themselves—complement C2 and complement C4—were correlated with these conditions. Another noteworthy association across both the transcriptomic and proteomic data was found with the *AGER* gene and its corresponding protein, the advanced glycosylation end-product-specific receptor.

Our comparative analysis with the Common Metabolic Disease Knowledge Portal revealed several overlaps in genetic markers associated with diabetic retinopathy. These findings affirm the potential significance of these markers. Notably, *IER3* (*p*-value = 0.015), *HLA-DQA2* (*p*-value = 1.06 × 10^−11^), C2 (*p*-value = 1.76 × 10^−10^), *VARS2* (*p*-value = 0.005), *VARS1* (*p*-value = 1.08 × 10^−7^), *BAG6* (*p*-value = 6.20 × 10^−13^), *TNXB* (*p*-value = 3.17 × 10^−10^), *AGER* (*p*-value = 2.09 × 10^−7^), glutamate receptor 4 (*p*-value = 0.016), and MHC class I polypeptide-related sequence B (*p*-value = 1.02 × 10^−8^) demonstrated significant associations within the CMDKP, reinforcing their potential relevance in diabetic retinopathy.

### 2.2. Gene Ontology PANTHER Analysis

In the analysis of genes uniquely associated with diabetic retinopathy at an FDR-adjusted *p*-value threshold of <0.05, we identified several notable findings. A total of nine genes uniquely associated with DR were found that are involved in the adaptive immune response process: *TRAV27*, *TRAV8-1*, *C2*, *TRAV20*, *TRAV2*, *TRAV12-1*, *TRBV10-3*, *HLA-DRB5*, and *TRBV2* (FDR 6.67 × 10^−5^). As well, *BAG6* and *AGER* are associated with the broader immune system process in addition to the previous ones listed (FDR 1.42 × 10^−2^). *VARS2* and *VARS1* are associated with valyl-TRNA aminoacylation (FDR 2.79 × 10^−2^). When the threshold was reduced to <0.005, only *VARS2* was found to be associated with valyl-TRNA aminoacylation (FDR 1.99 × 10^−2^).

In the combined analysis of genes unique to DR and those in common with diabetes mellitus at the threshold of <0.05, we identified *C2*, *C4A*, *CD8A*, *HLA-DRB5*, and *HLA-DQA2*, which have been associated with lymphocyte-mediated immunity (FDR 2.96 × 10^−2^). Moreover, a total of 22 genes were found that are also associated with the immune system process: *TRAV3*, *C2*, *TRAV17*, *TRBV6-1*, *TRBV18*, *TRAV2*, *TRAV12-1*, *TRAV12-2*, *TRAV23DV6*, *TRBV10-3*, *CD8A*, *C4A*, *HLA-DRB5*, *HLA-DQA2*, *TRAV27*, *TRAV8-1*, *AGER*, *BAG6*, *TRIM10*, *TRAV20*, *RIOK3*, and *TRVB2* (FDR 1.49 × 10^−9^). In addition to the previous ones, *IER3*, *HSPA1B*, *JUNB*, *DCLRE1B*, *ADGRG1*, *TRBV15*, and *FLOT1* were identified, which have been found to be associated with immune response to stimulus (FDR 9.58 × 10^−4^). Reducing the threshold to <0.005, only *TRAV3*, *C2*, *TRAV17*, *TRBV6-1*, *TRBV18*, *TRAV2*, *TRAV12-1*, *TRAV12-2*, *TRAV23DV6*, *TRBV10-3*, *CD8A*, *C4A*, *HLA-DRB5*, and *HLA-DQA2* were identified as being associated with adaptative immune response (FDR 1.20 × 10^−11^) and the immune system process (FDR 6.01 × 10^−5^). Finally, *C2*, *C4A*, *HLA-DRB5*, and *HLA-DQA2* were identified as being associated with B-cell-mediated immunity (FDR 1.30 × 10^−2^). *HLA-DQB1-AS1* and *SAPCD1-AS1* could not be mapped in GO PANTHER.

### 2.3. STRING Database Analysis

The STRING database analysis of genes associated with DR and in common with DM showed that *C2*, *JUNB*, *RIOK3*, *HLA-DQA2*, *HLA-DRB5*, *AGER*, *HSPA1B*, *BAG6*, *CD8A*, *C4A*, and *TRIM10* are linked with the immune system process (Gene Ontology) with an FDR of 0.0198. Moreover, *VARS2* and *VARS1* are associated with valyl-tRNA aminoacylation (FDR 0.0197) (Figure 2).

*TRAV23DV6*, *TRBV18*, *TRAV3*, *TRBV6-1*, *TRAV12-2*, *TRAV17*, *TRBV15*, *TRAV2*, *TRAV12-1*, *TRBV10-3*, *TRAV27*, *TRBV2*, *TRAV8-1*, *TRAV20*, *HLA-DQB1-AS1*, and *SAPCD1-AS1* were not present in the STRING database.

Protein analysis using the STRING database identified proteins involved in a number of biological processes with known or potential pathogenicity in DR/DM. Notably, the immune response emerged prominently; 13 proteins were identified that are significantly associated with this process: ubiquitin-like protein ISG15, cAMP-specific 3′,5′-cyclic phosphodiesterase 4D, T-cell surface protein tactile, interleukin-21, MHC class I polypeptide-related sequence B, interleukin-36 alpha, C-C motif chemokine 19, C-C motif chemokine 22, complement C4, advanced glycosylation end-product-specific receptor, complement C2, GRO-beta, and coiled-coil domain-containing protein 134, with an FDR of 0.00043. Our analysis also identified C-C motif chemokine 19, C-C motif chemokine 22, GRO-beta, interleukin-36 alpha, and interleukin-21 as being concurrently linked to cytokine activity molecular function with an FDR of 0.0143. Furthermore, the STRING database analysis reported the association between complement C2 and complement C4 with the activation of C3 and C5 based on Reactome pathways (FDR of 0.0152) (Figure 3).

Extensive STRING database analysis results are reported in the Appendix A (Appendix A).

In summary, the statistical over-representation test analysis using GO PANTHER and the protein pathway analysis using the STRING database revealed a significant association of the identified proteins and genes with several biological processes, particularly those related to the immune system. The key processes represented included the adaptive immune response, the immunoglobulin-mediated immune response, valyl-tRNA aminoacylation, and complement activation.

## 3. Discussion

Our study analyzed potential transcriptomic and proteomic markers for diabetic retinopathy (DR) using the OmicsPred PheWAS application. We identified several genes and proteins uniquely associated with DR, as well some shared with diabetes mellitus (DM). Findings from our study underscore the multifaceted associations of DR, potentially allowing further insight into its pathophysiology. Known pathways and components of the immune system emerged as prominently associated with DR, supporting previous studies [2,8,9,10,11,12]. Both the adaptive and immunoglobulin-mediated immune responses were identified. Multiple T-cell receptor-coding genes were associated; such findings align with prior studies, which have suggested an integral role of immune response, particularly adaptive immunity, in DR [9].

The complement system, a crucial part of the immune response, appears to be important as well, since both transcriptomic and proteomic data showed associations between DR and DM and key genes involved in this system, such as *C2* and *C4A* and their respective proteins (complement C2 and complement C4). The complement system has been implicated in the pathogenesis of DR in previous studies, with evidence suggesting its activation in diabetic retinopathy [12,13,14,15]. Our results are consistent with these findings, further strengthening the evidence for the role of the complement system in diabetic retinopathy.

Valyl tRNA aminoacylation emerged as a potential pathway associated with diabetic retinopathy; the altered expression of *VARS2*, a gene that encodes mitochondrial aminoacyl-tRNA synthetase [16], may suggest dysfunction within the mitochondria. These findings align with the current body of research that underscores the significant role of mitochondria in the pathophysiology of diabetic retinopathy [17,18,19].

Finally, the *AGER* gene and the advanced glycosylation end-product-specific receptor were associated with diabetic retinopathy at both the transcriptomic and proteomic levels. This observation aligns with prior studies which have suggested the implications of advanced glycation end products in the pathogenesis of diabetic retinopathy, underlying their contribution to diabetes vascular modifications, oxidative stress, and inflammation [18,20,21,22,23]. Moreover, a recent study suggested a relationship between the complement system and AGER [24]. The researchers found that, in hyperglycemic conditions, there was an upregulation of the complement component C3 in astrocytes. This process may be mediated by the activation of p38MAPK/NF, which is triggered by AGER signaling. Further investigations using NF-κB inhibitors, p38MAPK, and AGER inhibitors confirmed this observation, leading to a significant downregulation of C3. This evidence strengthens the link between AGER signaling and complement activation in the context of hyperglycemia, emphasizing their potential role in diabetes-associated complications.

Prior candidate gene association studies have evaluated the genetic associations with the development and progression of diabetic retinopathy. A range of genes have been implicated in DR, including, but not limited to, vascular endothelial growth factor (*VEGF*), advanced glycation end-product receptor (*AGER*), aldose reductase (*AKR1B1*), glucose transporter 1 (*GLUT1*), erythropoietin (*EPO*), transcription factor 7-like 2 (*TCF7L2*), and intercellular adhesion molecule 1 (*ICAM1*). Candidate gene studies have highlighted potential genes of interest involved in various biological processes, such as glucose metabolism, inflammation, angiogenesis, and neurogenesis. However, reproducibility has proven to be a challenge with many of these findings, and those that have been successfully replicated tend to display only weak genetic associations [25,26,27].

Numerous Genome-Wide Association Studies (GWASs) have been conducted. GWASs offer the advantage of an unbiased approach, enabling researchers to discover disease pathways that were previously unsuspected, scanning genetic variants across entire genomes to identify potential associations with DR. However, the results from these studies have often yielded conflicting and inconsistent findings [25]. These findings are well summarized in a recent comprehensive meta-analysis including 14 GWAS studies [28].

Furthermore, previous transcriptomic and proteomic studies have probed the complex biological underpinnings of diabetic retinopathy. These studies have identified diverse biological processes and pathways implicated in DR, and several potential biomarkers have emerged; the critical roles of immune response, inflammation, and angiogenesis in DR’s pathogenesis have been underscored in recent studies [29,30]. The complement system, advanced glycation end-product receptor (AGER), and numerous components of the immune system have all been suggested to play a role in the development and progression of DR [29,30]. Our research further supports, by association, the roles of AGER, the complement system, and the immune system in DR, and aligns with the existing body of research supporting these pivotal biological components as potential biomarkers.

Our findings are consistent with an animal study focusing on diabetic and nondiabetic rhesus monkeys, evaluating the role of inflammation in diabetic retinopathy [31]. Their investigation revealed a significantly higher expression of inflammatory mediators such as ICAM-1, TNF-α, VEGF, CRP, and MCP-1 in diabetic monkeys. These markers were predominantly located in the retinal and choroidal blood vessels and were associated with the severity of diabetes. The study’s immunohistochemical findings, which showcased an increased expression of these inflammatory markers in diabetic monkey eyes, resonate with our results and further support the hypothesis that local inflammation could be a central pathway leading to the onset and progression of diabetic retinopathy. In light of our in silico work, these monkey eyes are now being re-evaluated to look at complement and other inflammatory markers in the retina and choroid.

There are several potential limitations of this analysis. First, the data utilized in this study are derived from the OmicsPred database. For this reason, our study is subject to the same inherent limitations associated with this resource. Though this database has been shown to be reliable for other diseases [4], it is inherently dependent upon the validity of the initial input data. Additionally, we focused on data from the OmicsPred PheWAS application, which is based on a specific cohort from the UK Biobank (UKB). This could limit the generalizability of our findings, as the study population consists primarily of individuals of European descent [4,5]. Furthermore, it is worth noting that the data utilized in this study do not consider the varying degrees of severity of diabetic retinopathy or the type of diabetes mellitus. As well, our analyses document only an association, not a pathogenic role in DM/DR for the identified genes and proteins.

In conclusion, despite the limitations of our study, our findings contribute to a growing body of evidence implicating the involvement of the immune system, the complement system, mitochondrial function, T-cell receptors, and advanced glycosylation end-product receptor in DR. Our results may provide valuable insights into DR’s molecular landscape and highlight potential biomarkers for further investigation. Future studies should focus on validating the identified biomarkers in larger cohorts and exploring their potential roles in the pathogenesis and pathobiology of DR, and eventually in its treatment.

## 4. Materials and Methods

### 4.1. Data Sources

The OmicsPred PheWAS database (https://www.omicspred.org/Applications, accessed on 3 May 2023) [4] was queried to analyze transcriptomic and proteomic data and identify potential gene and protein biomarkers associated with diabetic retinopathy. We focused on two Phecodes—diabetic retinopathy and diabetes mellitus—to determine the genes and proteins common and unique to these conditions. We selected associated genes and proteins based on their false discovery rate (FDR)-adjusted *p*-values, with an FDR-adjusted *p*-value threshold of <0.05. Additionally, we identified those with a higher stringency threshold of <0.005 for further evaluation. In cases of commonality with different FDR-adjusted *p*-values, the most conservative FDR-adjusted *p*-value with the highest value was used. Genes not coding for proteins were noted but not included in further analyses. To further validate and expand upon our findings, we compared our identified transcriptomic and proteomic markers from OmicsPred with gene-level associations from the Common Metabolic Disease Knowledge Portal (CMDKP). (https://md.hugeamp.org/, accessed on 14 April 2024) [32]. The phenotype page of CMDKP offers “bottom-line” meta-analyzed variant associations and gene-level associations for specific phenotypes, incorporating aggregated results from various contributing datasets. We specifically analyzed the diabetic retinopathy phenotype under the ‘Common Variant’ tab, which presents gene-level associations calculated using the MAGMA (Multi-marker Analysis of GenoMic Annotation) method [32,33] (https://md.hugeamp.org/phenotype.html?phenotype=DiabeticRetino, accessed on 14 April 2024).

This comparison aimed to validate our data against GWAS studies and enhance the credibility of our identified markers for diabetic retinopathy.

### 4.2. Gene Ontology PANTHER 

We performed pathway analysis using the Gene Ontology PANTHER (Protein ANalysis THrough Evolutionary Relationships) system to identify the biological processes involved (http://www.pantherdb.org/, accessed on 15 May 2023) [6]. We focused on the biological processes associated with the different sets of genes identified, using the statistical over-representation test known as GO biological process complete. The analyses were performed separately, as follows:Genes uniquely associated with diabetic retinopathy with an FDR-adjusted *p*-value threshold of <0.05.Genes uniquely associated with diabetic retinopathy with an FDR-adjusted *p*-value threshold of <0.005.Combined genes uniquely associated with diabetic retinopathy and those in common with diabetes mellitus, with an FDR-adjusted *p*-value threshold of <0.05.Combined genes uniquely associated with diabetic retinopathy and those in common with diabetes mellitus, with an FDR-adjusted *p*-value threshold of <0.005.

The statistical over-representation test, GO biological process complete, was performed on May 2023.

### 4.3. STRING Database

In addition to the gene ontology PANTHER analysis, we performed protein pathway and protein–protein interaction analysis using the STRING database (https://string-db.org/, accessed on 18 May 2023) [7]. The analysis was performed on combined proteins uniquely associated with diabetic retinopathy and those in common with diabetes mellitus, with an FDR-adjusted *p*-value threshold of <0.05. Moreover, the analysis of combined genes uniquely associated with DR and those in common with diabetes mellitus, with an FDR-adjusted *p*-value threshold of <0.05, was performed using the string-db. STRING database, version 11.5, accessed in May 2023.

## Figures and Tables

**Figure 1 ijms-25-05456-f001:**
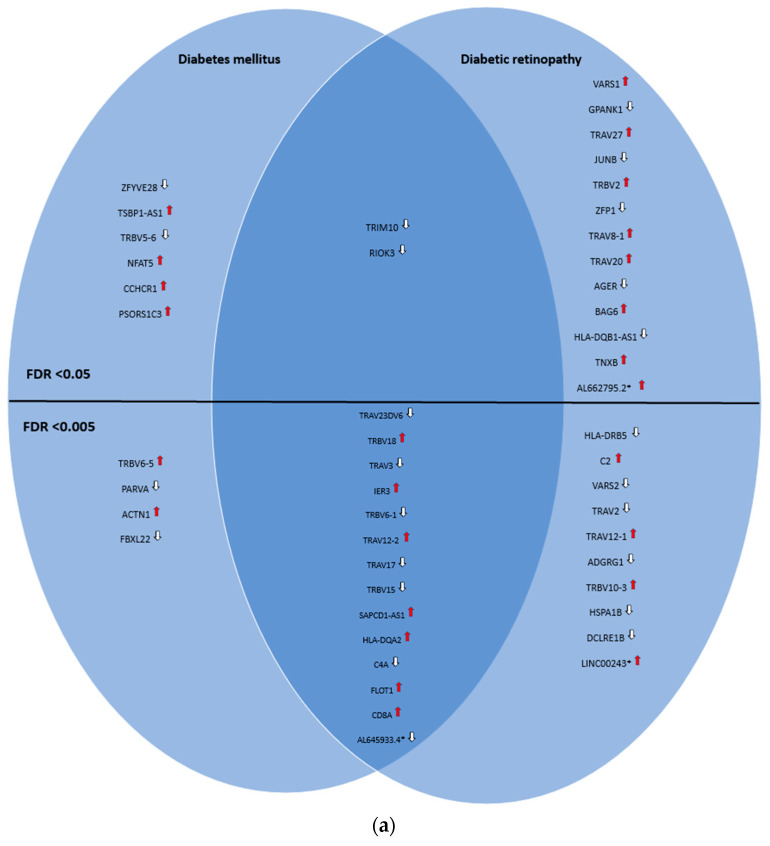
(**a**) Venn diagram illustrating the shared and unique genes associated with diabetic retinopathy and diabetes mellitus according to the OmicsPred PheWAS application database. The left circle represents the genes uniquely associated with diabetes mellitus. The right circle represents the genes uniquely associated with diabetic retinopathy. The area of overlap between circles represents genes that are associated with both of these conditions. In the upper portion, genes with an FDR-adjusted *p*-value < 0.05 are shown, while in the lower portion, genes with an FDR-adjusted *p*-value < 0.005 are shown. * Denotes pseudogenes or genes not coding for proteins (

 = hazard ratio > 1; 

 = hazard ratio < 1) (a comprehensive list of abbreviations can be found in Appendix A). (**b**) Venn diagram illustrating the shared and unique proteins associated with diabetic retinopathy and diabetes mellitus according to the OmicsPred PheWAS application database. The left circle represents the proteins uniquely associated with diabetes mellitus. The right circle represents the proteins uniquely associated with diabetic retinopathy. The area of overlap between circles represents genes that are common between these conditions. In the upper portion, genes with an FDR-adjusted *p*-value < 0.05 are shown, while in the lower portion, genes with an FDR-adjusted *p*-value < 0.005 are displayed (

 = hazard ratio > 1; 

 = hazard ratio < 1) (a comprehensive list of abbreviations can be found in Appendix A).

**Figure 2 ijms-25-05456-f002:**
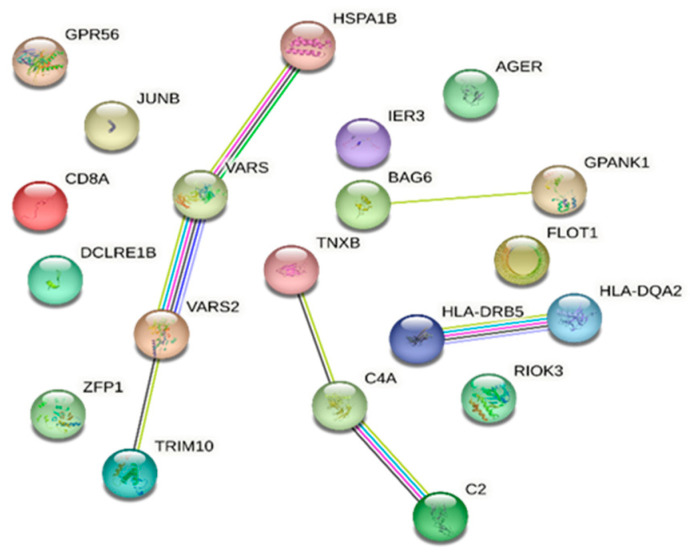
STRING database [7] network of combined genes associated with diabetic retinopathy and those in common with diabetes mellitus (a comprehensive list of abbreviations can be found in Appendix A).

**Figure 3 ijms-25-05456-f003:**
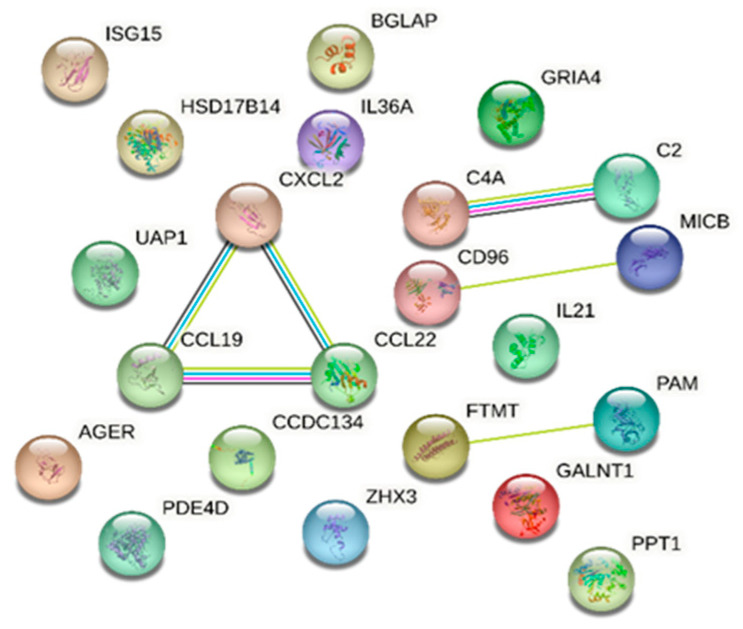
STRING database [7] network of combined proteins associated with diabetic retinopathy and those in common with diabetes mellitus (a comprehensive list of abbreviations can be found in Appendix A).

**Table 1 ijms-25-05456-t001:** (**a**) Proteomic and transcriptomic associations with diabetic retinopathy. (**b**) Proteomic and transcriptomic associations with diabetes mellitus. * Denotes pseudogenes or genes not coding for proteins (a comprehensive list of abbreviations can be found in Appendix A). Adapted from https://www.omicspred.org/Applications, accessed on 3 May 2023 [4].

(**a**)						
**Phenotype**	**Trait ID**	**Platform**	**Type**	**Description**	**Hazard Ratio|Effect Size**	**FDR-Adjusted *p*-Value**
250.7	HSD17B14.13972.4.3	SOMAscan	Protein	17-beta-hydroxysteroid dehydrogenase 14	0.86 (0.83 0.90)	3.11 × 10^−10^
250.7	UAP1.13580.2.3	SOMAscan	Protein	UDP-N-acetylhexosamine pyrophosphorylase	1.15 (1.11 1.20)	9.69 × 10^−10^
250.7	ISG15.14148.2.3	SOMAscan	Protein	Ubiquitin-like protein ISG15	1.15 (1.10 1.19)	1.92 × 10^−8^
250.7	CXCL3.CXCL2.2986.49.2	SOMAscan	Protein	Gro-beta/gamma	0.87 (0.84 0.91)	2.56 × 10^−7^
250.7	PDE4D.5255.22.3	SOMAscan	Protein	cAMP-specific3′,5′-cyclic phosphodiesterase 4D	0.89 (0.85 0.92)	4.79 × 10^−7^
250.7	IL21.7124.18.3	SOMAscan	Protein	Interleukin-21	0.90 (0.86 0.93)	1.27 × 10^−5^
250.7	GRIA4.10760.107.3	SOMAscan	Protein	Glutamate receptor 4	0.90 (0.86 0.93)	2.61 × 10^−5^
250.7	GALNT1.7090.17.3	SOMAscan	Protein	Polypeptide N-acetylgalactosaminyltransferase 1	1.11 (1.07 1.16)	2.78 × 10^−5^
250.7	C4A.C4B.4481.34.2	SOMAscan	Protein	Complement C4	0.90 (0.86 0.93)	3.51 × 10^−5^
250.7	C2.3186.2.2	SOMAscan	Protein	Complement C2	1.12 (1.08 1.17)	3.51 × 10^−5^
250.7	IL36A.14150.7.3	SOMAscan	Protein	Interleukin-36 alpha	0.90 (0.87 0.94)	4.37 × 10^−5^
250.7	CD96.9735.44.3	SOMAscan	Protein	T-cell surface protein tactile	0.90 (0.87 0.94)	0.0001568465824
250.7	BGLAP.11067.13.3	SOMAscan	Protein	Osteocalcin	1.11 (1.06 1.15)	0.0001657170479
250.7	FTMT.8048.9.3	SOMAscan	Protein	Ferritin, mitochondrial	0.91 (0.87 0.94)	0.0003308696342
250.7	CCDC134.5587.3.3	SOMAscan	Protein	Coiled-coil domain-containing protein 134	1.10 (1.06 1.15)	0.0003448553335
250.7	PPT1.9244.27.3	SOMAscan	Protein	Palmitoyl-protein thioesterase 1	1.09 (1.05 1.14)	0.00567148826
250.7	MICB.5102.55.3	SOMAscan	Protein	MHC class I polypeptide-related sequence B	0.92 (0.88 0.96)	0.007798151605
250.7	ZHX3.10036.201.3	SOMAscan	Protein	Zinc fingers and homeoboxes protein 3	0.92 (0.88 0.96)	0.01196628092
250.7	O00626	Olink	Protein	C-C motif chemokine 22	0.88 (0.85 0.91)	1.72 × 10^−7^
250.7	Q99731	Olink	Protein	C-C motif chemokine 19	0.89 (0.86 0.93)	2.37 × 10^−5^
250.7	Q15109	Olink	Protein	Advanced glycosylation end-product-specific receptor	0.90 (0.87 0.94)	7.44 × 10^−5^
250.7	ENSG00000211803	RNAseq	Gene expression	*TRAV23DV6*	0.84 (0.81 0.86)	5.45 × 10^−21^
250.7	ENSG00000211777	RNAseq	Gene expression	*TRAV3*	0.84 (0.81 0.88)	3.98 × 10^−14^
250.7	ENSG00000137331	RNAseq	Gene expression	*IER3*	1.17 (1.13 1.22)	2.38 × 10^−12^
250.7	ENSG00000211789	RNAseq	Gene expression	*TRAV12-2*	1.16 (1.12 1.21)	5.62 × 10^−11^
250.7	ENSG00000244731	RNAseq	Gene expression	*C4A*	0.86 (0.83 0.90)	7.92 × 10^−11^
250.7	ENSG00000276557	RNAseq	Gene expression	*TRBV18*	1.16 (1.11 1.20)	2.30 × 10^−9^
250.7	ENSG00000276819	RNAseq	Gene expression	*TRBV15*	0.87 (0.84 0.91)	7.39 × 10^−9^
250.7	ENSG00000137312	RNAseq	Gene expression	*FLOT1*	1.15 (1.10 1.19)	7.39 × 10^−9^
250.7	ENSG00000237541	RNAseq	Gene expression	*HLA-DQA2*	1.13 (1.09 1.18)	1.94 × 10^−8^
250.7	ENSG00000211797	RNAseq	Gene expression	*TRAV17*	0.88 (0.84 0.91)	3.78 × 10^−8^
250.7	ENSG00000286940	RNAseq	Gene expression	*AL645933.4* *	0.88 (0.85 0.92)	2.70 × 10^−7^
250.7	ENSG00000198502	RNAseq	Gene expression	*HLA-DRB5*	0.89 (0.85 0.93)	9.86 × 10^−6^
250.7	ENSG00000166278	RNAseq	Gene expression	*C2*	1.12 (1.08 1.17)	9.86 × 10^−6^
250.7	ENSG00000214894	RNAseq	Gene expression	*LINC00243* *	1.12 (1.07 1.16)	3.39 × 10^−5^
250.7	ENSG00000137411	RNAseq	Gene expression	*VARS2*	0.90 (0.87 0.94)	7.09 × 10^−5^
250.7	ENSG00000235663	RNAseq	Gene expression	*SAPCD1-AS1*	1.12 (1.07 1.17)	0.0001531493895
250.7	ENSG00000153563	RNAseq	Gene expression	*CD8A*	1.11 (1.07 1.15)	0.0001657170479
250.7	ENSG00000211776	RNAseq	Gene expression	*TRAV2*	0.90 (0.87 0.94)	0.0001657170479
250.7	ENSG00000211706	RNAseq	Gene expression	*TRBV6-1*	0.90 (0.87 0.94)	0.0002278762017
250.7	ENSG00000211785	RNAseq	Gene expression	*TRAV12-1*	1.11 (1.06 1.15)	0.0003050825425
250.7	ENSG00000205336	RNAseq	Gene expression	*ADGRG1*	0.91 (0.87 0.94)	0.0007282565197
250.7	ENSG00000204613	RNAseq	Gene expression	*TRIM10*	0.92 (0.88 0.95)	0.001918186381
250.7	ENSG00000275791	RNAseq	Gene expression	*TRBV10-3*	1.10 (1.06 1.15)	0.001918186381
250.7	ENSG00000204388	RNAseq	Gene expression	*HSPA1B*	0.91 (0.87 0.95)	0.002078760943
250.7	ENSG00000118655	RNAseq	Gene expression	*DCLRE1B*	0.92 (0.89 0.95)	0.002811484616
250.7	ENSG00000204394	RNAseq	Gene expression	*VARS1*	1.09 (1.05 1.13)	0.005775038565
250.7	ENSG00000204438	RNAseq	Gene expression	*GPANK1*	0.91 (0.87 0.95)	0.005984171578
250.7	ENSG00000211809	RNAseq	Gene expression	*TRAV27*	1.09 (1.05 1.13)	0.005984171578
250.7	ENSG00000171223	RNAseq	Gene expression	*JUNB*	0.92 (0.89 0.96)	0.007254641194
250.7	ENSG00000226660	RNAseq	Gene expression	*TRBV2*	1.09 (1.04 1.13)	0.01153288912
250.7	ENSG00000184517	RNAseq	Gene expression	*ZFP1*	0.92 (0.89 0.96)	0.016486343
250.7	ENSG00000211782	RNAseq	Gene expression	*TRAV8-1*	1.08 (1.04 1.13)	0.01747972125
250.7	ENSG00000211800	RNAseq	Gene expression	*TRAV20*	1.08 (1.04 1.12)	0.02578222421
250.7	ENSG00000101782	RNAseq	Gene expression	*RIOK3*	0.93 (0.89 0.96)	0.02900478075
250.7	ENSG00000280128	RNAseq	Gene expression	*AL662795.2* *	1.08 (1.04 1.13)	0.03022370348
250.7	ENSG00000204305	RNAseq	Gene expression	*AGER*	0.92 (0.89 0.96)	0.03214553862
250.7	ENSG00000204463	RNAseq	Gene expression	*BAG6*	1.08 (1.04 1.12)	0.03747012286
250.7	ENSG00000223534	RNAseq	Gene expression	*HLA-DQB1-AS1*	0.93 (0.89 0.96)	0.04255859
250.7	ENSG00000168477	RNAseq	Gene expression	*TNXB*	1.08 (1.03 1.12)	0.04789597686
(**b**)						
**Phenotype**	**Trait ID**	**Platform**	**Type**	**Description**	**Hazard Ratio|Effect Size**	**FDR-Adjusted *p*-Value**
250	HSD17B14.13972.4.3	SOMAscan	Protein	17-beta-hydroxysteroid dehydrogenase 14	0.92 (0.89 0.95)	1.28 × 10^−5^
250	CXCL3.CXCL2.2986.49.2	SOMAscan	Protein	Gro-beta/gamma	0.92 (0.89 0.95)	2.25 × 10^−5^
250	UAP1.13580.2.3	SOMAscan	Protein	UDP-N-acetylhexosamine pyrophosphorylase	1.08 (1.05 1.11)	6.48 × 10^−5^
250	C2.3186.2.2	SOMAscan	Protein	Complement C2	1.08 (1.05 1.11)	0.0002108515966
250	PAM.5620.13.3	SOMAscan	Protein	Peptidyl-glycine alpha-amidating monooxygenase	0.93 (0.91 0.96)	0.001096113192
250	BGLAP.11067.13.3	SOMAscan	Protein	Osteocalcin	1.07 (1.04 1.10)	0.001163693677
250	GALNT1.7090.17.3	SOMAscan	Protein	Polypeptide N-acetylgalactosaminyltransferase 1	1.06 (1.03 1.09)	0.01402345692
250	ISG15.14148.2.3	SOMAscan	Protein	Ubiquitin-like protein ISG15	1.06 (1.03 1.09)	0.02371665877
250	IL21.7124.18.3	SOMAscan	Protein	Interleukin-21	0.94 (0.92 0.97)	0.02824020531
250	PDE4D.5255.22.3	SOMAscan	Protein	cAMP-specific 3′,5′-cyclic phosphodiesterase 4D	0.95 (0.92 0.97)	0.03541793772
250	Q15109	Olink	Protein	Advanced glycosylation end-product-specific receptor	0.93 (0.91 0.96)	0.00051925312
250	Q99731	Olink	Protein	C-C motif chemokine 19	0.94 (0.92 0.97)	0.03734158925
250	ENSG00000211803	RNAseq	Gene expression	*TRAV23DV6*	0.89 (0.87 0.91)	3.73 × 10^−15^
250	ENSG00000276557	RNAseq	Gene expression	*TRBV18*	1.12 (1.09 1.15)	1.98 × 10^−10^
250	ENSG00000211777	RNAseq	Gene expression	*TRAV3*	0.91 (0.89 0.94)	1.39 × 10^−7^
250	ENSG00000137331	RNAseq	Gene expression	*IER3*	1.10 (1.07 1.13)	2.44 × 10^−7^
250	ENSG00000211706	RNAseq	Gene expression	*TRBV6-1*	0.91 (0.89 0.94)	1.30 × 10^−6^
250	ENSG00000211789	RNAseq	Gene expression	*TRAV12-2*	1.09 (1.06 1.12)	5.82 × 10^−6^
250	ENSG00000211797	RNAseq	Gene expression	*TRAV17*	0.92 (0.90 0.95)	2.58 × 10^−5^
250	ENSG00000235663	RNAseq	Gene expression	*SAPCD1-AS1*	1.09 (1.06 1.13)	2.72 × 10^−5^
250	ENSG00000276819	RNAseq	Gene expression	*TRBV15*	0.92 (0.90 0.95)	3.80 × 10^−5^
250	ENSG00000237541	RNAseq	Gene expression	*HLA-DQA2*	1.08 (1.05 1.11)	5.68 × 10^−5^
250	ENSG00000286940	RNAseq	Gene expression	*AL645933.4* *	0.93 (0.90 0.95)	8.72 × 10^−5^
250	ENSG00000244731	RNAseq	Gene expression	*C4A*	0.93 (0.90 0.95)	0.0001277312243
250	ENSG00000211721	RNAseq	Gene expression	*TRBV6-5*	1.08 (1.05 1.11)	0.000269472771
250	ENSG00000137312	RNAseq	Gene expression	*FLOT1*	1.07 (1.04 1.10)	0.0009346706263
250	ENSG00000197702	RNAseq	Gene expression	*PARVA*	0.94 (0.91 0.96)	0.003054863328
250	ENSG00000072110	RNAseq	Gene expression	*ACTN1*	1.07 (1.04 1.10)	0.003371476599
250	ENSG00000153563	RNAseq	Gene expression	*CD8A*	1.07 (1.04 1.10)	0.003371476599
250	ENSG00000197361	RNAseq	Gene expression	*FBXL22*	0.94 (0.91 0.97)	0.004496292136
250	ENSG00000159733	RNAseq	Gene expression	*ZFYVE28*	0.94 (0.91 0.97)	0.007293018296
250	ENSG00000204613	RNAseq	Gene expression	*TRIM10*	0.94 (0.92 0.97)	0.01259977175
250	ENSG00000101782	RNAseq	Gene expression	*RIOK3*	0.94 (0.92 0.97)	0.01402345692
250	ENSG00000225914	RNAseq	Gene expression	*TSBP1-AS1*	1.06 (1.03 1.09)	0.02250145314
250	ENSG00000211728	RNAseq	Gene expression	*TRBV5-6*	0.95 (0.92 0.97)	0.02496127518
250	ENSG00000102908	RNAseq	Gene expression	*NFAT5*	1.06 (1.03 1.09)	0.02802043893
250	ENSG00000204536	RNAseq	Gene expression	*CCHCR1*	1.06 (1.03 1.09)	0.03367545913
250	ENSG00000204528	RNAseq	Gene expression	*PSORS1C3*	1.06 (1.03 1.09)	0.03733127066

## Data Availability

Publicly available datasets were analyzed in this study. These data can be found here: https://www.omicspred.org/Applications (accessed 3 May 2023).

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
