# Peer review of "Exploring the Associated Genetic Causes of Diabetic Retinopathy as a Model of Inflammation in Retinal Diseases"

_ijms, 2024, doi:10.3390/ijms25105456_

Round 1

Reviewer 1 Report

Comments and Suggestions for Authors

The authors have conducted an investigation into potential biomarkers and biological processes associated with diabetic retinopathy (DR) utilizing transcriptomic and proteomic data. These findings present avenues for future research into novel diagnostic and therapeutic strategies for DR. Below are suggestions aimed at enhancing the current study:

1.       Figure 1 could be enhanced by displaying the upregulated and downregulated genes/proteins distinctly, facilitating clearer interpretation.

2.       To improve readability, authors may consider illustrating potential pathways associated with the findings, aiding in comprehension for readers.

3.       It would be beneficial to ascertain whether the identified targets have been studied by other research groups. If so, referencing additional studies could augment the manuscript's comprehensiveness and credibility.

4.       Authors may enhance the depth and breadth of analysis by incorporating a wider range of data sources, thereby enriching the content and providing a more comprehensive understanding of the investigated phenomena.

Author Response

Dear Reviewer,

Thank you for your constructive feedback on our manuscript. We have taken your suggestions into consideration and made the following adjustments to enhance our study's comprehensiveness and readability.

To provide clearer interpretation, we have updated Figure 1 to distinctly display the upregulated and downregulated genes/proteins. We used arrows to indicate whether the hazard ratio was greater than or less than 1, allowing for better visual clarity.

To ascertain whether the identified targets have been studied by other research groups, we compared our transcriptomic and proteomic data from Omicspred with gene-level associations from the Common Metabolic Disease Knowledge Portal (CMDKP). The Phenotype page of CMDKP provides meta-analyzed variant associations and gene-level associations for specific phenotypes. We specifically examined the diabetic retinopathy phenotype under the 'Common Variant' tab, calculated using the MAGMA (Multi-marker Analysis of GenoMic Annotation) method. This comparison aimed to validate our findings against broader GWAS studies. 

In our methods section we added the following:

To further validate and expand upon our findings, we compared our identified transcriptomic and proteomic markers from Omicspred with gene-level associations from the Common Metabolic Disease Knowledge Portal (CMDKP). (https://md.hugeamp.org/)

The Phenotype page of CMDKP offers "bottom-line" meta-analyzed variant associations and gene-level associations for specific phenotypes, incorporating aggregated results from various contributing datasets. We specifically analyzed the diabetic retinopathy phenotype under the 'Common Variant' tab, which presents gene-level associations calculated using the MAGMA (Multi-marker Analysis of GenoMic Annotation) method. (https://md.hugeamp.org/phenotype.html?phenotype=DiabeticRetino). This comparison aimed to validate our data against GWAS studies and enhance the credibility of our identified markers in diabetic retinopathy.

and referenced the website and its publication.

In our results section we added the following:

Our comparative analysis with the Common Metabolic Disease Knowledge Portal revealed several overlaps in genetic markers associated with diabetic retinopathy. These findings affirm the potential significance of these markers. Notably, IER3 (p-value = 0.015), HLA-DQA2 (p-value = 1.06e-11), C2 (p-value = 1.76e-10), VARS2 (p-value = 0.005), VARS1 (p-value = 1.08e-7), BAG6 (p-value = 6.20e-13), TNXB (p-value = 3.17e-10), AGER (p-value = 2.09e-7), Glutamate Receptor 4 (p-value = 0.016) and MHC Class I Polypeptide-Related Sequence B (p-value=1.02e-8) demonstrated significant associations within the CMDKP, reinforcing their potential relevance in diabetic retinopathy.

Thank you again for your valuable suggestions, and we look forward to your continued feedback.

Best regards

Reviewer 2 Report

Comments and Suggestions for Authors

Ijms-2916773

Title: Exploring the associated genetic causes of diabetic retinopathy as a model of inflammation in retinal disease

Authors used the OmicsPred database to identify genes and proteins associated DR and DM. In addition, biological processes and protein interactions related to the identified biomarkers by using Ontology PANTHER and STRING database. Authors identified the associated biomarkers of the immune response, complement system, advanced glycation end product formation and specific receptors, and mitochondrial function with DR pathophysiology.

Results of the study are not so new but may be useful for further studies to validate the roles of the identified biomarkers in the pathogenesis of DR. However, several issues are still existed in the manuscript.

Major biomarkers including VEGF, ICAM-I, VCAM-I, and IL-6 were not identified in this method. Please explain the reasons of missing major biomarkers in the list. Similar to GWAS, the results from this method may yield conflicting and inconsistent findings in the future.

This method can only identify upregulated or increased biomarkers in DM and DR. However, downregulated factors such as PEDF are also important for the pathogenesis of DM and DR. Thus, this method can see only one aspect of the pathogenesis of DM and DR. It is a limitation of this method. Authors should discuss the weak point of this method.

There are a lot of abbreviations without full description. Authors must prepare the table to show all abbreviations and full description appeared in the manuscript. Furthermore, in the figure legends (figs. 2 and 3), all full description of the abbreviations should be described respectively.

End of comments

Author Response

Dear Reviewer,

Thank you for your insightful feedback on our manuscript. We've carefully reviewed your comments and taken steps to address the issues you've raised:

To improve the readability of Figure 1, we've added arrows to distinguish between upregulated and downregulated genes/proteins, indicating whether the hazard ratio was greater than or less than 1.

To enrich the analysis, we compared our transcriptomic and proteomic data from Omicspred with gene-level associations from the Common Metabolic Disease Knowledge Portal (CMDKP). The Phenotype page of CMDKP provides meta-analyzed variant associations and gene-level associations for specific phenotypes. We specifically examined the diabetic retinopathy phenotype under the 'Common Variant' tab, calculated using the MAGMA (Multi-marker Analysis of GenoMic Annotation) method. This comparison aimed to validate our findings against broader GWAS studies. 

To address the use of abbreviations, we've created a comprehensive table in the supplemental materials. In our manuscript, we've added the following reference: "A comprehensive list of abbreviations can be found in Supplementary Table S3." 

We appreciate your feedback. Thank you for your constructive comments, and we look forward to your continued review.

Best Regards

Round 2

Reviewer 1 Report

Comments and Suggestions for Authors

The authors have provided answers to prior comments and enhanced the manuscript. 

Reviewer 2 Report

Comments and Suggestions for Authors

Although it is not so new, this manuscript may include some useful information for clinicians and researchers.